# Secretory Structures of *Pogostemon auricularius*: Morphology, Development, and Histochemistry

**Jiansheng Guo [1] and Cheng Zhou [2],\*** 

1   School of Medicine, Zhejiang University, Hangzhou 310058, China; jsguo518@zju.edu.cn
2   Key Laboratory of Bio-organic Fertilizer Creation, Ministry of Agriculture, Anhui Science and Technology University, Bengbu 233100, China
\*   Correspondence: zhouch@ahstu.edu.cn; Tel.: +86-0550-6732024

**Abstract:** *Pogostemon auricularius*, an aromatic plant in Lamiaceae, has wide application in pharmaceutical preparations. However, little is known about the secretory structures that contain the medicinal compounds. In this study, two kinds of glandular trichome types, including peltate glandular trichomes and short-stalked capitate trichomes, were identified in the leaves and stems by cryo-scanning electron microscope. Oil secretion from the glands contained lipids, flavones, and terpenes, and the progresses of secretion were different in the two glands types. The investigation by transmission electron microscope indicated that the endoplasmic reticulum system and plastids were involved in the biosynthesis of oils in the two glandular trichomes. The vacuoles showed a new role in the oil preparations and storage. The synthesized oil could be transported from the head cell to the sub-cuticular space by different way in the two glands. Comparative analysis of the development, distribution, histochemistry and ultrastructures of the secretory structures in *Pogostemon auricularius* led us to propose that the two glands may make different contribution to the collection of medicinal compounds. Furthermore, the characteristics of two glands in the secretory stage probably indicated the synthesizing site of metabolite.

**Keywords:** *Pogostemon auricularius*; peltate glandular trichomes; short-stalked capitate trichomes

## 1. Introduction

Medicinal plants are important for the healthcare system of a key portion of world population. Some medicinal plants in Lamiaceae contain compounds that can be used for the treatment of diabetes, cancer, and depressive disorder [1]. Previous studies on Lamiaceae had shown that some secretory structures, including peltate trichomes and capitate trichomes, were responsible for the production of secondary metabolite [2,3]. *Pogostemon auricularius (P. auricularius)* is the most important tropical and aromatic crop in Labiatae, which is cultured primarily in Southeast Asia, India, and Brazil [4]. As a medicinal plant, the leaf and stem of *P. auricularius* are used in the treatment of cat fever. The EtOH extract exhibits spasmolytic activity on preliminary screening [5]. The plant derived compounds, which included terpene and some derivatives of diterpene acid, may be the effective constituents identified [6]. However, little is known about the secretory structures that contain the medicinal compounds. Information about the total composition of the secreted material, the secretory process, and synthesizing site of metabolite are lacking.

In this study, we described for the first time the secretory structure of *P. auricularius* regarding the development, distribution, histochemistry, and secretory process. Our results indicated the possible synthesizing site of metabolite and secretory process.

## 2. Materials and Methods

### 2.1. Plant Materials

*P. auricularius* was cultured in a growth chamber with a photoperiod of 16 h (light)/8 h(dark) at 22/18 °C. The relative humidity of the growth chamber ranged from 50% to 70%. Leaves and stems at different developmental stages were separated for analyzing the glandular trichomes.

### 2.2. Scanning Electron Microscopy (SEM) and Transmission Electron Microscopy (TEM)

For cryo-SEM, fresh leaves and stems at different developmental stages were subjected to a series of treatments, including liquid nitrogen frozen, sublimated, and gold-coated in Quorum PP2000T Cryo-SEM system. Samples were investigated with a Hitachi S-3400N scanning electron microscope. We investigated 30 peltate trichomes and 30 short-stalked capitate trichome of *P. auricularius*. For transmission electron microscopy, the harvested samples were initially fixed in 2.5% glutaraldehyde solution, and were then fixed in 1% osmium tetroxide. Subsequently, these samples were subjected to a series of dehydration from 30% to 100%, and they were finally embedded in Eponate 12 resin. The embedded samples were cut to a thickness of 70 nm for the observation of transmission electron microscope. The number of glandular trichomes at each developmental stage is 20.

### 2.3. Light Microscopy

The development and histochemistry of glandular trichomes were examined by light microscopy. We examined 20 glandular trichomes for each trichomes type at different staining tests. Different histochemical tests were applied to examine the primary classes of metabolites of glandular trichomes, and the observations were made by an Olympus microscope. Localization of total lipids was investigated using Neutral red (Sigma) and Sudan black B (Sigma). Unsaturated lipids were analyzed using osmium tetroxide (Ted Pella). Naturstoffreagent A (Aladdin) was used for detecting the flavonoids (under UV365 emission LP397). Periodic acid-Schiff (PAS) reagent (Sigma) and Sudan III (Sigma) was applied to detect the localization of polysaccharide and lipids, respectively. The localization of terpenes and pectins was examined using Nadi reagent and ruthenium red (Sigma).

## 3. Results

### 3.1. Morphology, Distribution and Histochemistry of Secretory Structures

The surface view of stems (Figure 1A) and leafstalks (Figure 1B) showed numerous glandular trichomes and bristle hairs. The structure was also found at the adaxial surface (Figure 1C) and abaxial surface (Figure 1D) of leaves. According to the morphology of glandular trichomes, we observed the peltate trichomes (Figure 1E) and short-stalked capitate trichomes (Figure 1F). Each of the two trichome types of *P. auricularius* were easily characterized.

Peltate trichome with a length of 30 μm (± 7) and a diameter of 50 μm (± 11) distributed on the leaves and stems. The cryo-SEM micrographs showed the secretory process of peltate glandular trichomes (Figure 2A–C). In the early secretory stage, the cuticle was not found at the apex of the peltate trichome (Figure 2A). As secretion progressed, the cuticle was prominent outside of the head cell because of the accumulation of secretion oil in the sub-cuticular space (SCS) (Figure 2B). In the post-secretory stage, the secretion came into the environment and the cuticle contacted with the secretory cell tightly (Figure 2C). The composition of the secretory compounds differed depending on the stage of development. In young and underdeveloped trichomes, different reactions for lipids, polysaccharides, and terpenoids material were much weaker than mature trichomes. The mature trichomes showed a well-developed head region with a SCS, which was filled with essential oil. In the peltate glandular trichomes, an intense violet or blue-violet staining was observed in the secretion contained in the SCS by the observation of terpenoids material using the Nadi reaction (Figure 2D). The reaction with ruthenium Red (Figure 2E) and PAS (Figure 2H), staining for polysaccharides,

was clearly positive in mature peltate trichomes. Yellow staining for flavones with Naturstoff reagent A was intense, which stated that there were so many flavones in the peltate trichomes (Figure 2G). The histochemical tests that were carried out in mature peltate trichomes to detect total lipid gave positive results. Gold-yellow secondary fluorescence was observed by the Neutral Red staining and under UV light, indicating much lipid in the peltate trichomes (Figure 2K). These results were also confirmed using $OsO_4$ (Figure 2F) for unsaturated lipids, Sudan Black B (Figure 2I), and Sudan III (Figure 2J) for total lipid.

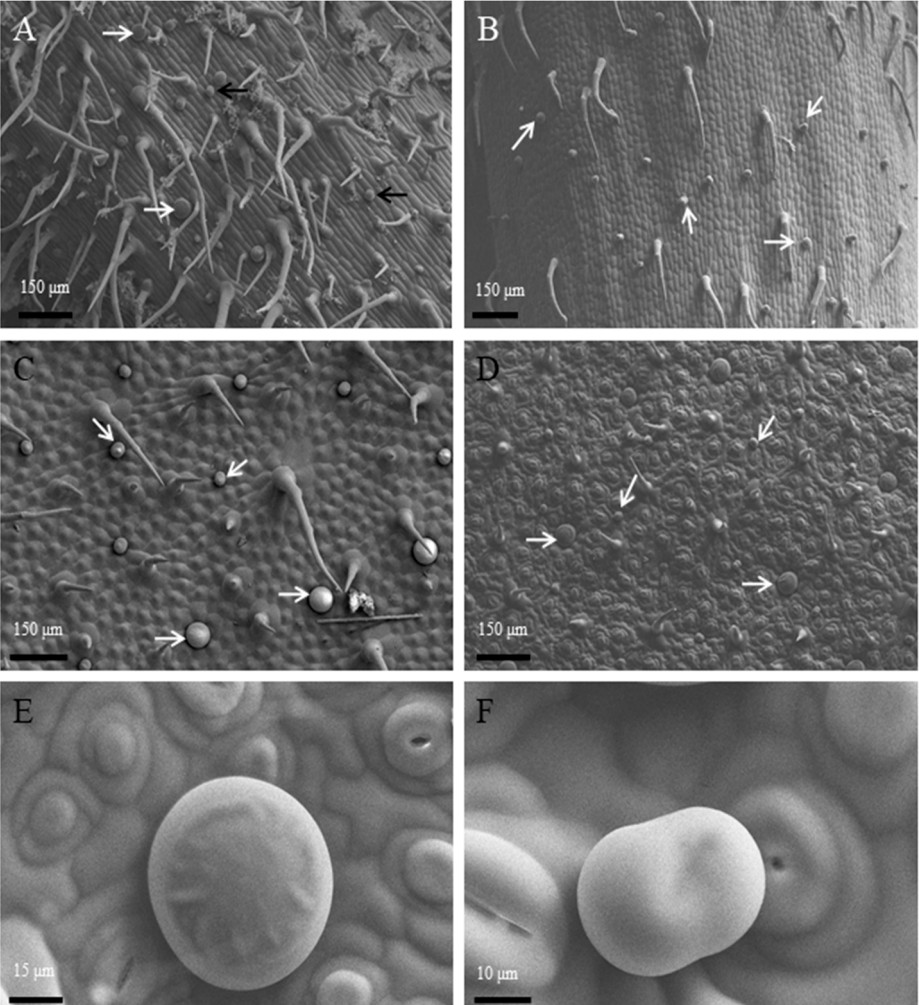

**Figure 1.** Morphology and localization of glandular trichomes on the surface of *P. auricularius*. (**A–F**) (Cryo–SEM): The observations of stems (**A**) and leafstalks (**B**) containing peltate glandular trichomes (white arrows) and short-stalked capitate glandular trichomes (black arrows) among non-glandular trichomes. Adaxial surface (**C**) and abaxial surface (**D**) view with two types of trichomes in different phases. Higher magnification of (**D**) maturely peltate glandular trichomes (**E**) and short-stalked capitate glandular trichomes (**F**).

Short-stalked capitate trichome with a length of 30 μm (± 6) and 40 (± 10) μm was found on both leaf sides and stems. In contrast to the peltate trichome, the secretory process of this trichome type was quite obvious. In the developing trichome, one slight gap was found between the two head cells (Figure 3A). This process started when the trichome was turgescent: one big oil droplet (Figure 3B) and sometimes two oil droplets (Figure 3C) appeared outside the SCS following the oil that oozed through the cuticle. The oil droplet was not observed by traditional SEM because of the dehydration treatment to sample (not shown). The small oil droplet was also shown by light

microscopy (Figure 3F). The light microscopy showed the result of histochemical investigations (Figure 3D–K). In the short-stalked capitate trichomes, an intense violet or blue-violet staining was observed in the secretion that was contained in the SCS by the Nadi staining of terpenoids material (Figure 3D). The results of PAS (Figure 3E) and ruthenium Red (Figure 3H) staining suggested that polysaccharides existed in the secretion products of this glandular trichome type. In the secretory stage of the head cells, mature capitate trichome was stained, showing fluorescent yellow-orange with the Naturstoff reagent A, which indicated flavones (Figure 3G). The secretion of short-stalked capitate trichomes stained positively for lipophilic: staining with $OsO_2$ (Figure 3F), Sudan Black B (Figure 3I), and Sudan III (Figure 3J) for lipids revealed lipids. These results were further confirmed using Neutral Red under UV light (Figure 3K).

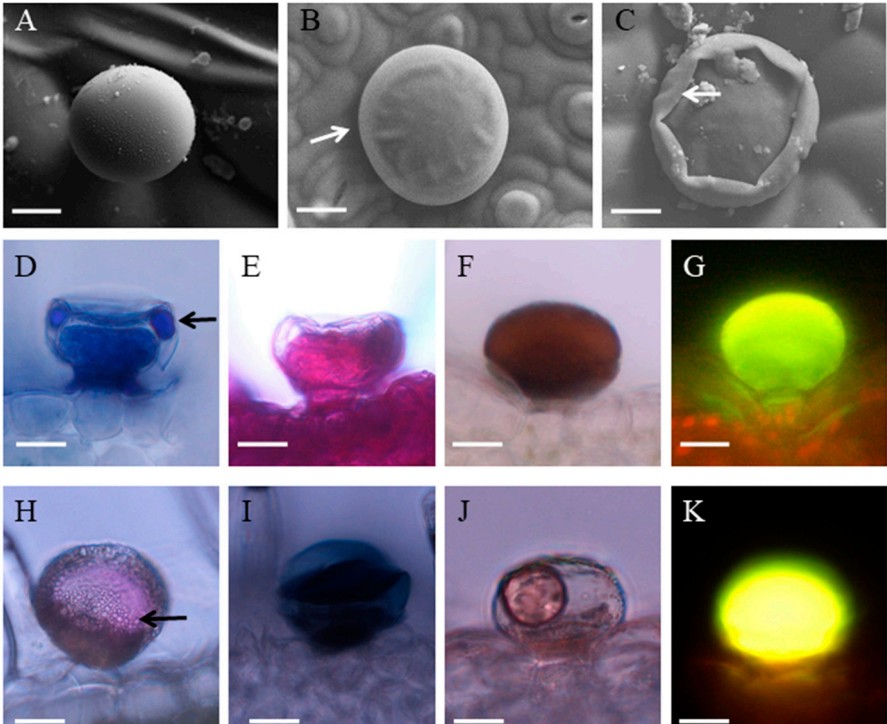

**Figure 2.** Histochemistry and secretion of peltate glandular trichomes in *P. auricularius*. (**A–C**) Cryo–SEM micrographs showing the progress of secretion: (**A**) The developing trichome without thickened cuticle; (**B**) secretory stage with protruding cuticle (arrow); (**C**) the collapse of the sub-cuticular space (arrow) after secretion. (**D–K**) Histochemistry of the peltate glandular trichomes: (**D**) the Nadi staining for terpenes revealed that the essential oil droplets (arrow) existed in the sub-cuticular space; (**E**) Ruthenium Red test showing the trichome stained red; (**F**) the OsO4 staining for the head cell and narrow stalk cell; (**G**) the apical cells react positively with Naturstoff reagent A; (**H**) the oil droplets (arrow) within the sub-cuticular space stained red in the PAS test for polysaccharides; The staining for total lipids with Sudan Black B (**I**), Sudan III (**J**), and Neutral red (**K**) suggests the accumulation of total lipids in the peltate glandular trichomes. Bar = 20 μm.

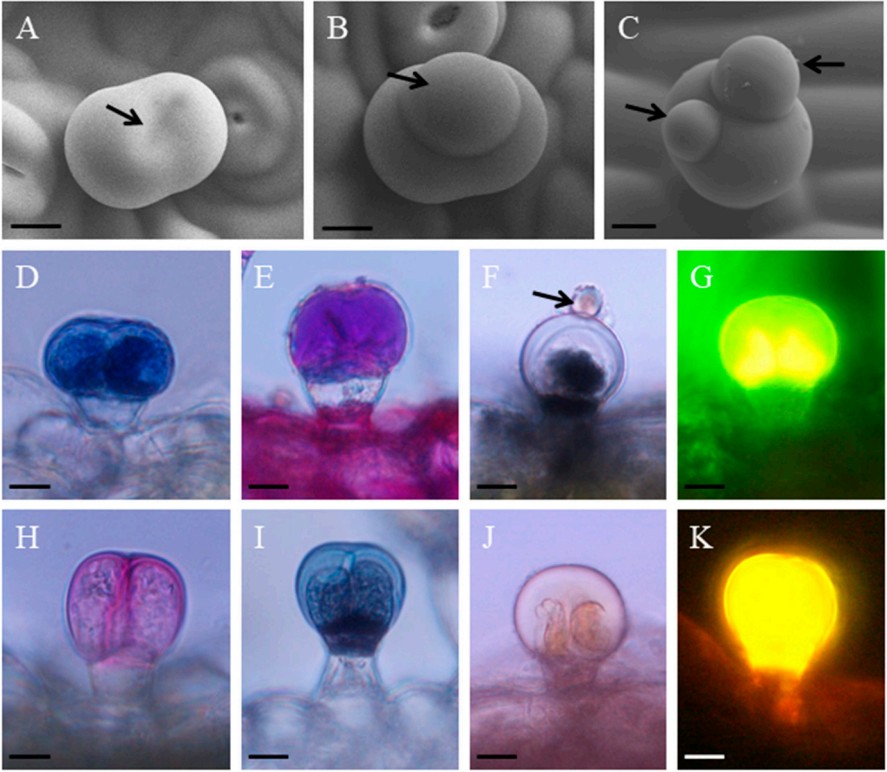

**Figure 3.** The histochemistry and secretion of short-stalked capitate glandular trichomes in *P. auricularius.* (**A–C**) Cryo–SEM micrographs showing the process of secretion: (**A**) the pre-secretory stage with the secretory center (arrow); (**B**) the initial stage of the secretory release (arrow); (**C**) the secretory droplets are getting bigger and the number increase to two (arrows). (**D–K**) histochemistry of the short-stalked capitate glandular trichomes: (**D**) the Nadi staining for terpenes is positive in the apical cells; (**E**) PAS test for polysaccharides in apical cells; (**F**) the OsO4 staining for the apical cells and the droplet (arrow) stained black; (**G**) the staining for the apical cells using Naturstoff reagent A; (**H**) Ruthenium Red test showing the apical cells stained red; (**I**) the staining for the apical cells using Sudan Black B and the stalk cell stained black; (**J**) positive staining reaction with Sudan III; and, (**K**) gold-yellow secondary fluorescence observed with Neutral Red under UV light. Bar = 10 μm.

*3.2. Ultrastructural Aspects of Secretory Structures*

During two glandular trichomes development, three different stages, including presecretory, secretory, and postsecretory were observed. Peltate glandular trichomes in the presecretory stage had one big head cell, one narrow stalk cell, and one basal cell. The head cell and stalk cell were ultrastructurally similar to the meristematic cells, showing numerous ribosomes and proplastids, few small vacuoles, and large nuclei and nucleoli, while basal cells remained vacuolated and they contained fewer organelles (Figure 4A). The ER was relatively sparse and the proplastids were small at early stages, but the glands became larger, rough endoplasmic reticulum and Golgi became more abundant (Figure 4B), and plastids with starch grain contained inclusion with high electron density in the stroma and tubular membranes (Figure 4C). In the presecretory stage, the cuticle in close connect with the ell wall was very thin and had lower electron density than typical cuticle (Figure 4D). The way in which the SCS is formed had been uncertain. The vesicle-like structures within the SCS may suggest that vesicle-like structures carry deposits of cutin to the expanding cuticle (Figure 4E, arrow). The plastids in the stalk cell had a small prolamellar body-like region of crystalloid plastid membranes and an isolated membrane (Figure 4F). A marked separation of the thickened cuticle was found in the initial phase of secretion across the apical surface of the glandular trichome and the secretory oil filling in the SCS (Figure 4G). At this stage, the smooth endoplasmic reticulum (SER)

occurred in greater profusion and the mature plasmids were more abundant (Figure 4H). Many oil droplets appeared between the cytomembrane and cell wall (Figure 4G,H). The plastids with a few starch grains were surrounded by the long segments of SRE and they were in close contact with small vacuoles and oil droplets (Figure 4H,I). Interestingly, some oil droplets were half surrounded by small vacuoles (Figure 4I) and some other oil were found in the single large central vacuole of head cell (Figure 4G). During the postsecretory stage, the SCS was filled with lipid droplets and the cuticle at the apex of peltate glandular trichome appeared to be markedly thicker than that covering the lateral sides of the gland (Figure 4J). The single large central vacuole got much bigger and was filled with some electron-opaque material (Figure 4J,L). Many oil droplets accumulated in cytoplasm and SER appeared like a concentric ring (Figure 4K). Sometimes, the dilapidated head cell with one large central vacuole had little cytoplasm (Figure 4L).

Short-stalked capitate trichomes had two head cells, one stalk cell and one vacuolated basal cell. Similar to peltate glandular trichomes, two head cells of short-stalked capitate trichomes in the presecretory stage had a dense cytoplasm, large nuclei, few small vacuoles, numerous ribosomes, abundant mitochondria, and proplastids (Figure 5A–C). Proplastids contained inclusion with high electron density in the stroma and tubular membranes (Figure 5B) and few vesicles were found near Golgi (Figure 5C). The thick cuticle covering glandular head cells were penetrated by a random fibrillar network (Figure 5D). The secreted material may be released via a cuticular fibrillar network. In the secretory stage, the short-stalked capitate trichomes had one smaller SCS that was filled with lipid droplets and the thinner cuticle that was not thickened at the apex of the dome as compared with peltate glandular trichomes (Figure 5E,F). There was a significant difference in the proliferation of the endomembrane system between the presecretory and secretory stages (Figure 5G). The dominant component was RER in the glandular cells. In the parietal cytoplasm, long and narrow cisternae and forming stacks were parallel to each other and to the plasma membrane (Figure 5G,I). In addition, numerous small vacuoles, mature plastids without starch grain, and abandent Golgi in secretory cells were also the remarkable feature of short-stalked capitate trichomes in the secretory stage (Figure 5H). In the head cell, numerous Golgi vesicles were found and were close to plasma membrane (Figure 5J). The Golgi stacks commonly assembling in groups occurred in the cytoplasmic regions that were rich in both RER elements and polyribosomes (Figure 5G,I). Similar to peltate glandular trichomes, small vacuoles and plastids were surrounded by rough endoplasmic reticulum (RER) tightly (Figure 5H). Big vacuoles and numerous RER were close to plasmodesmata between secretory cells and the stalk cell, which indicated that the head cells might be nourished by stalk cell (Figure 5I). In the postsecretory stage, two head cells containing one single large central vacuole were dilapidated and the SCS was larger than ever because of the exudation of secretions (Figure 5K). Nucleus and cytoplasm were squeezed to the edge of the head cell (Figure 5L).

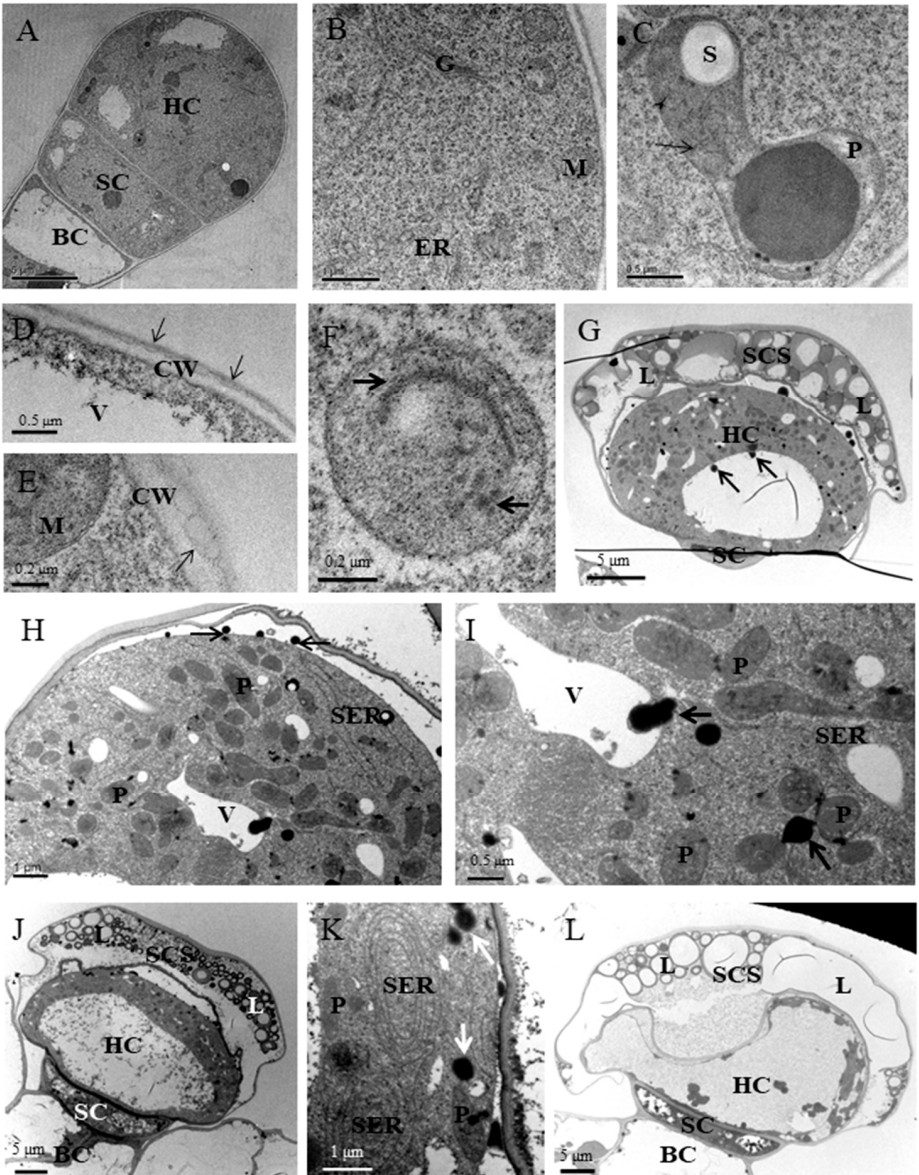

**Figure 4.** Ultrastructural aspects of peltate glandular trichomes in *P. auricularius*. (**A–F**) Presecretory stage: (**A**) Transverse section through a presecretory peltate glandular trichome with one cytoplasmically dense head cell. BC, basal cell; SC, stalk cell; HC, head cell; (**B**) the head cell appears typical characteristics of developing cells containing abundant ribosomes, Golgi (G), mitochondria (M) and few endoplasmic reticulum (ER); (**C**) amoeboid proplastids (P) with a few starch grains (S) contain numerous tubular membranes (arrow); (**D**) the cuticular (arrow) initial outside the apical cell; CW, cell wall; V, vacuoles; (**E**) vesicle-like structures (arrow) close to the outside of cell wall; (**F**) typical stalk cell plastid with a small prolamellar body-like region in the crystalloid plastid membranes (right arrow) and an isolated membrane (left arrow). (**G–I**) Secretory stage: (**G**) peltate glandular trichomes in secretory stage with one cytoplasmically dense head cell and the sub-cuticular space (SCS) filled with plentiful electron-light lipid deposits (L); (**H**) the higher magnification of (**G**) showing abundant plastids (P) with a few starch grains and small vacuoles (V) in close contact with the long segments of smooth endoplasmic reticulum (SER). Many oil droplets (arrows) appeared between cytomembrane and cell wall. (**I**) the mature head cell containing electron-opaque material (arrows) in association with vacuoles and plastids. (**J–L**) Post-secretory stage: (**J**) one single large central vacuole appeared in head cell and plentiful electron-light lipid droplets in the sub-cuticular space (SCS); (**K**) many oil droplets accumulated in cytoplasm and SER appeared like concentric ring; and, (**L**) the dilapidated head cell with one large central vacuole and few cytoplasm.

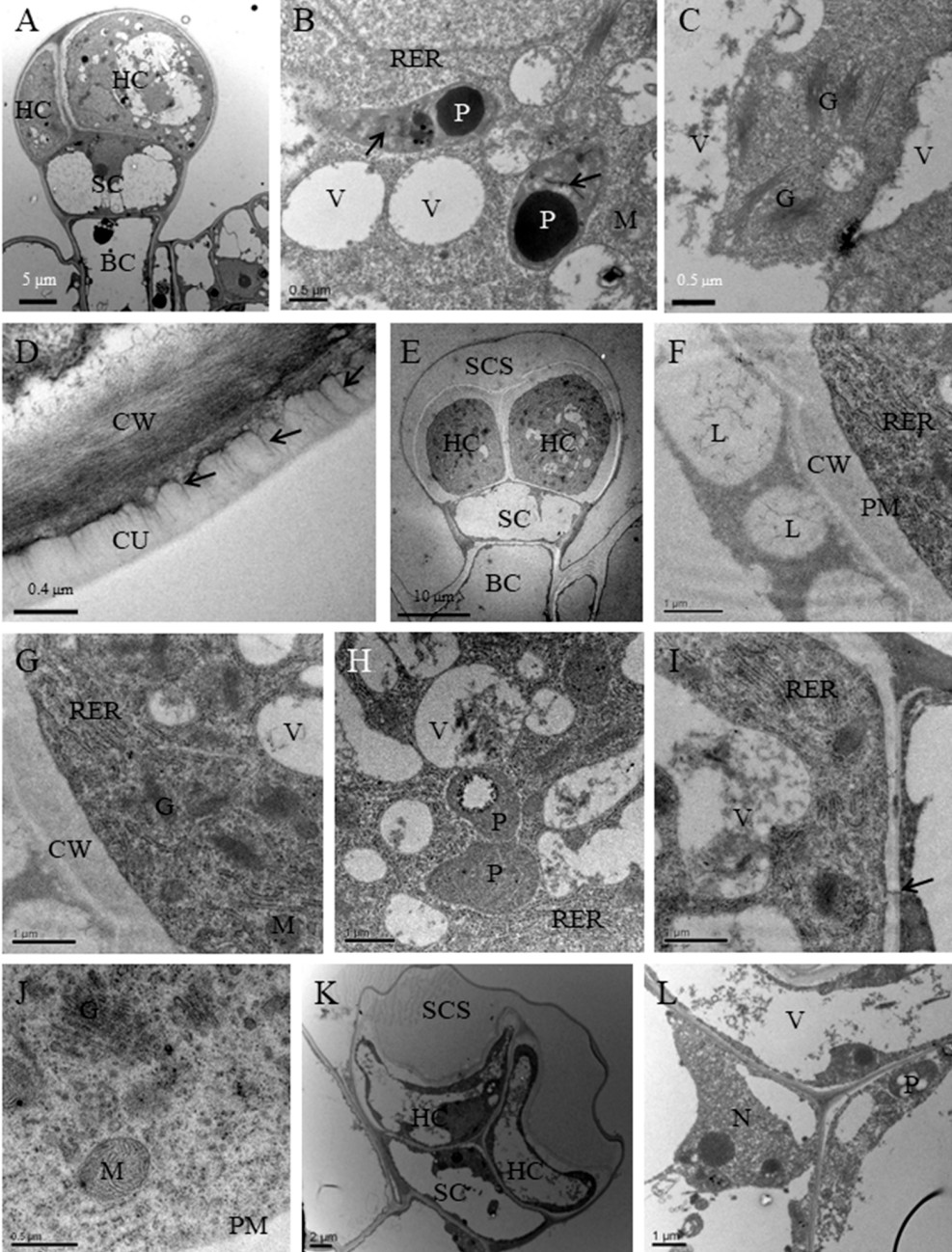

**Figure 5.** Ultrastructural aspects of short-stalked capitate glandular trichomes in *P. auricularius*. (**A–D**) Presecretory stage: (**A**) the two head cells occurred in developing trichome (HC), one stalk cell (SC) and one basal cell (BC) but the sub-cuticular space (SCS); (**B**) the head cell contained abundant roughendoplasmic reticulum (RER) close to small vacuoles (V) and plastids (P) which contain numerous tubular membranes (arrows); (**C**) a large number of Golgi (G) without obvious secretory vesicles; (**D**) a random fibrillar network (arrows) on the thin reticulate cuticle (CU) is evident. (**E–J**) Secretory stage: (**E**) the trichomes in the secretory stage contained two head cell (HC), a narrow stalk cell (SC) and a basal cell (BC) and the sub-cuticular space (SCS); (**F**) the sub-cuticular space filled with large lipid droplets (L); CW, cell wall; (**G**) numerous Golgi and plastids in close contact with RER; (**H**) portion of head cell showing plastids and small vacuoles are surrounded by RER; (**I**) portion of head cell showing big vacuoles and numerous RER close to plasmodesmata (arrow) which connects secretory cells and stalk cell; (**J**) numerous Golgi with many vesicles near the plasma membrane (PM). (**K–L**) Post-secretory stage: (**K**) two dilapidated head cells contain one single large central vacuole and the SCS without secretions; (**L**) nucleus (N) and cytoplasm are squeezed to the edge of the head cell.

## 4. Discussion

### 4.1. Analysis in Morphology, Development and Histochemistry

The remarkable characteristic feature of Lamiaceae species is the presence of numerous bristle hairs, peltate, and capitate glandular trichomes occuring abundantly in epidermis. All of these organs possess an independently characteristic morphology, ontogeny, histochemistry, and secretion process. There are more detailed descriptions of trichomes that are available in some reports for many commercially important genera, e.g., *Plantaginaceae* [7], *Origanum* [8], and *Lippiascaberrima* [9]. A previous study on *P. auricularius* was focused on the medicinal compounds and little was known about the secretory structures that were responsible for the secondary metabolism biosynthesis.

Observations on *P. auricularius* showed peltate trichomes and short-stalked capitate trichomes that were distributed on leaves, leaf stalk, and stems. In previous studies, the peltate glandular trichomes were described in several different morphology and contained different numbers of secretory cells in the head region [10–13]. The peltate glandular trichomes of *P. auricularius* had only one big secretory cell in the apical region, although the size was similar to that of previous studies. The short-stalked capitate trichomes had been described in *Plectranthus ornatus* [14], *Siderites syriaca* [15], *Leonotis leonurus* [10], and *Cucurbita pepo subsp. pepovar. styriaca* [16]. However, our study confirmed that the diversity of the glandular trichomes had not well classified and understood. The short-stalked capitate trichome corresponding to the capitate type I described by Werker et al. [12] consisted of one basal cell, one stalk cell, and two-celled head region. The capitate trichomes in Labiatae were the most common and they were found in virtually all the species studied. Observations on the development of the two glands types showed that they all originated from a single epidermis cell that was similar to glandular trichomes in other Lamiaceae species. Various color reactions were useful to study the localization and composition of the secreted material. The histochemical results indicated that there were terpenes, polysaccharides, flavones, and lipids in the head region of glands. Monoterpenes are the main components of the secretory material of the Lamiaceae [17] and Rutaceae species [2]. Sesquiterpenes and other terpenes were dominant in the essential oil of the Asteraceae [18]. Terpenoids can protect the plant from destruction by herbivores and other pathogens or attract pollinating insects to flowers [19]. The abundant terpenoids in SCS of two mature gland types of *P. auricularius* may affect the alimentary system of herbivores, thereby protecting plants from the destruction by the herbivores. There were lipophilic components in the secretion of the fully developed capitate glandular trichomes. These results conform to the result in other species of Lamiaceae [20]. As assumed in previous studies, the bulk of the essential oils were found in the peltate trichomes of Lamiaceae [21,22], the secretory material that was produced by peltate trichomes of *P. auricularius* contained lipids. In plants, the flavones may function as sunscreen pigments and/or antioxidants to alleviate adverse effects that are caused by UV light or oxidative damage, and to regulate auxin transport [23,24].

### 4.2. Site of Secondary Metabolism Biosynthesis

Some taxonomically distant plants share similar ultrastructural features of gland secretory cells, displaying relatively few Golgi, extensive endoplasmic reticulum, abundance mitochondria, and dense cytoplasm [24,25]. Previous studies have indicated that the possible sites of lipid accumulation were SER [26], vacuoles [27], cytoplasm [28], and the combination of plastids and SER [29]. In the secretory stage, the mature peltate plastids had plastoglobuli and an isolated membrane enclosing a large plastoglobule. These structures were reported to be related with oil biosynthesis. Although the lipid-containing plastids was not found in the glandular cells of short-stalked capitate trichomes, indicating the important role of plastids in the secondary metabolism, the close relationships between the secretory process and plastid chnages may imply that the plastids are involved in the biosynthesis of lipids and monoterpene. In addition, other evidence, including the biosynthesis of unmodified monoterpenes by isolated leucoplasts, the immunocytochemical localization of the limonene synthase in Peppermint secretory cell leucoplasts, and the documented origin of monoterpenes, which is

supposed to be localized in plastids [30–32], indicate that monoterpene synthesis is initiated within leucoplasts.

Abundant SER was a common feature of lipid-secreting glandular cells [27]. Based on the obvious changes of ER during the secretory process in lipophilic glands, previous investigators have supposed that ER may participate in the biosynthesis, accumulation, and secretion of terpenes [33,34]. In this study, the dominant, long and narrow RER that surrounds the plastids and Golgi in secreting short-stalked capitate trichomes of *P. auricularius* should not be ignored. However, young capitate trichomes had few RER. The RER rich in short-stalked capitate trichomes in the secretory phase, the presence of osmiophilic material within the RER, and the tight association with plasma membrane and plastids suggested that RER had a role in secondary metabolite biosynthesis. When compared with short-stalked capitate trichomes, the peltate trichomesin secretory phase had much longer SER, which was close to oil droplets and plasmids. The abundant Golgi bodies in the secreting short-stalked capitate trichomes are in line with previous studies, in which Golgi bodies are responsible for the biosynthesis of polysaccharides in the secretory trichomes [35]. The function of Golgi has been confirmed in capitate glandular trichomes of other species [26,27]. The secreting short-stalked capitate trichomes of *P. auricularius* had abundant Golgi with many secretory vesicles, while peltate trichomes contained much less Golgi bodies. The exuberant activity of Golgi may indicate its important role in secondary metabolite process and secretion.

Amelunxen [27] has assumed that the terpenoids were synthesized in the vacuoles. In the secretory cell of peltate trichomes of *P. auricularius*, electron-opaque material and lipid-like material that ocurred in small vacuoles may indicate that the vacuoles were involved in the biosynthesis of lipids. Additionally, the vacuoles were exhausted anomalously and the extraplasmatic space was enlarged during the secretion of lipid-like material out of small vacuoles. Thus, these observations further indicated the possible role of vacuoles in the biosynthesis of lipids. However, no further evidence was observed for confirming the hypothesis. Previous researchers have supposed that the vacuoles may not generate, but only process, the secretory material [36,37]. The different behavior of Golgi, ER system, and vacuoles in peltate trichomes and short-stalked capitate trichomes of *P. auricularius* suggested that the two gland types might have different sites of secondary metabolism biosynthesis. In addition, the distinctive organelles in the secretory stage may be the taxonomic character of the two glandular trichomes types.

### 4.3. Possible Secretory Mechanisms

For so long, the secretory process of oil, including transport from biosynthesis site to plasma membrane, transport through plasma membrane and cell wall to SCS, and release from SCS to environment attracted investigators. Some data indicated that secretory material was transported by direct ER-plasma membrane connections [38], based on the distribution of lipid deposits within the ER, and the position of ER with plastids and plasma membrane. Multivesicular bodies and paramural bodies may be crucial for transportng synthesized material [39]. The connection of the ER system and plasma membrane in peltate trichomes and short-stalked capitate trichomes of *P. auricularius* suggested that the transport of secretory material from ER to the cell wall might be through the connection of ER membranes with the plasmalemma. The abundant secretory oil in the periplasmic space and peripheral cytoplasm of peltate trichomes suggested that the oil might not be transported through multivesicular bodies or paramural bodies.

Numerous Golgi stacks with many vesicles in the capitate trichomes of secretory phase were found to be associated with RER and plasma membrane. The apparent structural polarity of Golgi stacks in the secretory cells of short-stalked capitate trichomes probably reflects the functional characteristic in the export of secondary metabolite. The close spatial association of Golgi stacks with RER indicates that RER participates in the biosynthesis of the protein component of the secretory product, and Golgi may be involved in the biosynthesis of the polysaccharide component, as supposed in the capitate trichomes of *Leonotis leonurus* by [11]. The transport of the secretion of trichomes from head cells to the

periplasmic space was through the connection of Golgi vesicles with plasma membrane. A previous study in other trichomes suggested a similar mechanism of oil secretion [40]. In the secretory stage, the cuticle tuber had been observed in short-stalked capitate glandular trichomes of *P. auricularius* by cryo-SEM and light microscopy. The small oil droplet at the base of cuticle tuber showed the phenomenon of secretory release. The release process of oil in SEM is consistent with that reported for *Cucurbia pepo* var. styriaca [16]. In *Calceolaria adscendens*, several environmental factors, predator contact, temperature, or humidity, can result in the partial rupture of the cuticle in capitate trichome [41]. The secretion that was released into the environment by *Cucurbitapepo subsp. pepovar. styriaca* [16] has been known to happen through the cuticular pores in the apical trigon of long-stalked glandular trichome. However, neither the partial rupture of the cuticle nor cuticular pores had been observed in the short-stalked capitate trichomes of *P. auricularius*. It was possible that the cuticle of short-stalked capitate trichomes, with its random fibrillar network channels, could allow the release of some secretion components.

In contrast to short-stalked glandular trichomes, no oil droplets were observed outside the SCS of peltate glandular trichomes. In post secretory stage, the cuticle at the apex of the dome was about twice as thick as that covering the lateral sides of the gland. This phenomenon corresponds to that of peltate glandular trichomes of *Plectranthus ornatus* [14]. The absence of cuticle thickness in apical region suggests that there may be a different way between the two glandular trichomes types to release the essential oil. The peltate glandular trichomes may gradually release the essential oil into the environment through the gaps among the cutin accumulations. The possibility that the cuticle remains relatively thicker with age has been the subject of some debate. The source of cutin for forming the SCS of Lamiaceae glandular trichomes has also been uncertain. It was suggested that new cutin was added to the expanding cuticle of glandular trichomes in Peppermint [27] and Origanum [28]. In young peltate glandular trichomes of *P. auricularius*, the vesicle-like structures outside the cell wall indicated the possible role of these structures in the formation of cuticle, as proposed by Kim and Mahlberg [42]. The thickness of the cuticle was uniform in the early secretory stage. Nevertheless, the cuticle at the apex of the dome became much thicker than that covering the lateral sides of the gland. Thus, it appears that the cuticle is reinforced during expansion and it become thicker with age.

## 5. Conclusions

Peltate glandular trichomes and short-stalked capitate trichomes of *P. auricularius*, which contained terpenes, polysaccharides, flavones, and lipids, were the secretory structure responsible for the medicinal components. They may have different sites of secondary metabolism biosynthesis and secretory process, thinking about hat plasmids, ER, Golgi, and vacuoles presented different behavior in each development stage.

**Author Contributions:** Funding acquisition, J.G. and C.Z.; Methodology, J.G.; Supervision, C.Z.; Writing—original draft, J.G.; Writing—review & editing, J.G. and C.Z.

**Funding:** This research was funded by the Experimental Technology Research Project of Zhejiang University (SZD201603), the Natural Science Foundation of Zhejiang Province (LQ17C050001), the National Natural Science Foundation of China (31600210, 31370214 and 31301058) and the Natural Science Foundation of Education Department of Anhui province (KJ2018ZD051).

**Conflicts of Interest:** The authors declare no conflict of interest. The funders had no role in the design of the study; in the collection, analyses, or interpretation of data; in the writing of the manuscript, or in the decision to publish the results.

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
