# Peer review of "Secretory Structures of Pogostemon auricularius: Morphology, Development, and Histochemistry"

_symmetry, doi:10.3390/sym11010013_

Round 1
Reviewer 1 Report
The study of Guo and Zhou (symmetry-396990-peer-review-v1) investigates Pogostemon auricularius, an aromatic plant of the family Lamiaceae that has wide pharmaceutical use although little is known about the secretory structures that contain the medicinal compounds. Using cryo-scanning electron microscopy, two kinds of glandular trichome types including peltate glandular trichomes and short-stalked capitate trichomes were identified in the leaves and stems. Oil secretion from the glands contained lipids, flavones and terpenes and the progress of secretion were different in the two glands types. The investigation by transmission electron microscope indicated that the endoplasmic reticulum system and plastids were involved in the biosynthesis of oils in the two glandular trichomes. By an comparative analysis of the development, distribution, histochemistry and ultrastructures of the secretory structures in P. auricularius authors conclude that the two glands may make different contribution to the collection of medicinal compounds.
The manuscript is quite well written, most experiments are properly designed, and any structural and morphological knowledge of P. auricularius is important to better understand and make use of its medical potential.
However, two main issues in the manuscript require careful consideration and revision by the authors before I can recommend any publication.
1. The main issue is the lack of information about the numbers of biological replicates analysed in this study in order to ensure repeatability for other researchers.
2. In this context, it is essential to describe the histochemical stainings with more details (see line 62-65) including where chemicals were purchased from etc.
Minor:
- shorten discussion by avoiding mere repetition of results (e.g. line 276-79)
- references: if possible add some more recent studies, since only two papers cited are quite recent, whereas all others are 12 to over 30 years old
- English grammar and word check is required throughout MS (e.g. line 37,38)
Author Response
Response to Reviewer 1 Comments
Point 1: The main issue is the lack of information about the numbers of biological replicates analysed in this study in order to ensure repeatability for other researchers.
Response1: For cryo-SEM, we investigated 30 peltate trichomes and 30 short-stalked capitate trichome of Pogostemon auricularius. For TEM investigation, the number of glandular trichomes at each developmental stage is 20. For light microscopy, we examined 20 glandular trichomes for each trichomes type at different staining test. And we have added these infomation to the paper.
Point 2: In this context, it is essential to describe the histochemical stainings with more details (see line 62-65) including where chemicals were purchased from etc.
Response 2: Neutral red (Sigma) and Sudan black B (Sigma) were used to localize total lipids, osmium tetroxide (Ted Pella) for unsaturated lipids, Naturstoffreagent A (Aladdin) for detection of flavonoids (under UV365 emission LP397), periodic acid-Schiff (PAS) reagent (Sigma) for polysaccharides, Sudan III (Sigma) for lipids, NADI reagent (Sigma) for terpenes, ruthenium red (Sigma) for pectins. And we have added these infomation to the paper.
Minor:
1. Shorten discussion by avoiding mere repetition of results (e.g. line 276-79)
Response: The discussion was changed to " In the secretory stage, the mature peltate plastids had plastoglobuli and an isolated membrane enclosing a large plastoglobule."
2. references: if possible add some more recent studies, since only two papers cited are quite recent, whereas all others are 12 to over 30 years old
Response: We have added 4 recent studies about glandular trichomes of plant. The list as following:
1. Jacek, J; Agata K; BoĹĽena D. Micromorphological and histochemical attributes of flowers and floral reward in Linaria vulgaris (Plantaginaceae). Protoplasma. 2018, 255, 1763-1776.
2. Agata, K. Comparative micromorphology and anatomy of flowers and floral secretory structures in two Viburnum species. Protoplasma. 2017, 254, 523-537.
3. Wood, BW. Flavonoids, alkali earth, and rare earth elements affect pecan pollen germination. HortScience. 2017, 52, 85-88.
4. Nick, B; Stefan, B; Frank, S; Gerd, H; Alain, T. The development of type VI glandular trichomes in the cultivated tomato Solanum lycopersicum and a related wild species S. habrochaites. BMC Plant Biology. 2015, 15, 289.
3. English grammar and word check is required throughout MS (e.g. line 37,38)
Response: The statement was changed to " However, little is known about the secretory structures which contains the medicinal compounds. The information about the total composition of the secreted material, the secretory process and synthesizing site of metabolite are lacking."

Reviewer 2 Report
This manuscript describes in details the secretory structures present on the leaf and stem of Pogostemon auricularius.
Pogostemon auricularius is a plant which produces medical compounds and these secretory structures have an important role to play in this process. The histochemical work and morphological evidences provided in the manuscript are very elegant, however there are certain issues with the study that need to be addressed
1. However this study only details the morphological and the histochemical traits of these structures, and does not provide any molecular data to complement these findings.
2. Though a thorough understanding of the morphological characteristics of this secretory structures is important, but it doesn't connect how it will benefit our understanding of the applications of the medical compounds secreted by this plant. The manuscript would have benefited if the authors could give us the broader scope of this study and its scientific application.
3. The manuscript can be improved for English language
Author Response
1. However this study only details the morphological and the histochemical traits of these structures, and does not provide any molecular data to complement these findings.
Response 1: The Pogostemon auricularius is not the model plant and information for genome is lack. It is difficult to get the genes which play an important role in development of glandular trichomes and metabolite synthesis. Next, We will get the molecular data of these structures through gene sequencing.
2. Though a thorough understanding of the morphological characteristics of this secretory structures is important, but it doesn't connect how it will benefit our understanding of the applications of the medical compounds secreted by this plant. The manuscript would have benefited if the authors could give us the broader scope of this study and its scientific application.
Response 2: Our observations explain the morphologies and histochemistry of Pogostemon auricularius trichomes for metabolite storage, biosynthesis and release and provide a framework for further studies of these important metabolic cellular factories. This is required to better exploit their potential, in particular for the oil extraction processing and identifying important molecular in oil biosynthesis.
3. The manuscript can be improved for English language
Response 3: We have improved the English language and resubmited the manuscript.

Round 2
Reviewer 1 Report
The revision has improved the MS, which now can be accepted for publication in Symmetry.